# Improving Lane Detection Performance for Autonomous Vehicle Integrating Camera with Dual Light Sensors

**Yunhee Lee** [1], **Min-ki Park** [2] **and Manbok Park** [3,*]

1   Litbig, Seongnam-si 13487, Korea; leeyunis@gmail.com
2   Mando, Seongnam-si 13487, Korea; mk.park@halla.com
3   Department of Electronic Engineering, College of Convergence Technology, Korea National University of Transportation, Chungju-si 27469, Korea
*   Correspondence: ohnnuri@ut.ac.kr; Tel.: +82-43-841-5369

**Abstract:** Automotive companies have studied the development of lane support systems in order to secure the Euro New Car Assessment Program (NCAP)'s high score. A front camera module is applied with safety assistance systems in an intelligent vehicle. However, the front camera module has limitations in terms of backlight conditions, entering or exiting tunnels, and night driving because of lower image quality. In this paper, we propose an integrated camera with dual light sensor for improving lane detection performance under the worst conditions. We include a new algorithm to enhance image data quality and improve edge detection and lane tracking using illumination information. We evaluate the tests under various conditions on a real road. These tests are performed on 728 km of road (under various external situations and lane types) for false alarm rates. The experimental results show that the system is promising in terms of reliability, enhancement, and improvements.

**Keywords:** advanced driving assistance system (ADAS); front camera module; lane support system; lane departure warning system; lane keeping assistance system; dual light sensor; lane detection; illumination sensor; light intensity detection; image data quality enhancements

## 1. Introduction

In 2010, the eSafety Challenge Euro New Car Assessment Program (NCAP) announced Euro NCAP Advanced, which includes a new reward system for new safety technologies. Such a system provides a complement to NCAP's existing star rating scheme and rewards those manufacturers that promote new technologies with safety benefits. Automotive companies have studied the development of an advanced driving assistance system (ADAS) in order to secure NCAP's high score. As a result, safety assistance systems, such as lane departure warning, blind spot monitoring, attention assistance, and autonomous emergency braking, are offered by automotive companies as options on their newest models. In particular, lane support systems have increasingly become widespread as safety assistance systems. Therefore, since 2014, Euro NCAP has opted to include lane support systems as standard requirement [1].

Major automobile manufacturers have actively researched lane support systems that employ a front camera module. The front camera module, which incorporates a lane recognition algorithm, executes lane support functions, such as lane departure warning systems (LDWS) and lane keeping assistance systems (LKAS), to help improve driver safety [2,3]. However, the front camera module has limitations in terms of backlight conditions, entering or exiting tunnels, and night driving because of lower image quality. In order to overcome these limitations, many studies are conducted. Methods that remove noise from images by applying filtering are used [4,5], or methods that are adaptive to changes in lighting are used [6–8].

In this paper, in order to improve the lane detection algorithm for lane support systems in automotive applications, we propose an integrated camera with a dual light sensor that

can detect a lane under the worst conditions. In addition, integrating a dual solar sensor in the front camera module can reduce the size of the system and is cost-effective [9].

The rest of this paper is organized as follows. Related studies on the vision-based lane detection and light intensity detection for dual light sensor are summarized in Section 2. Section 3 presents the proposed method. Section 4 presents the experimental results, and the paper is concluded in Section 5.

## 2. Related Work

### 2.1. General Methods for Vision-Based Lane Detection

Lane detection can be divided into four parts: (1) feature extraction, (2) line detection, (3) line fitting, and (4) lane tracking [3]. Lane features such as edges are extracted from the input image. Mu et al. and Tu et al. extracted lane features for lane detection using a Sobel filter [10,11]. Wu et al. and Wang et al. extracted lane features using a Canny edge detector [12,13]. Fang et al., Yim et al., and Jung et al. extracted lane features using line filters [14–16], and Bertozzi et al. extracted features using region filters [17]. From the extracted features, line detection selects lane features and removes non-lane features. Aung et al. selected lane features using the Hough transform [18]. Lee selected lane features using an edge distribution function [19]. Jung et al. selected lane features using the Hough transform and an edge distribution function [20]. Taubel et al. and Borkar et al. selected lane features using inverse perspective mapping [21,22]. Line fitting creates a mathematical model using selected lane features. Zhao et al. made a model using spline [23]. Obradovic et al. made a model using fuzzy line and fuzzy point [24]. Lin et al. made a straight line model [25]. Wu et al. and Mu et al. made a model using linear parabolic [10,12]. Lane tracking predicts the position in the next frame and sets the ROI(Region Of Interest). Borkar et al., Wu et al., and Obradovic et al. tracked the lane using a Kalman filter [13,24,26]. Kim et al. and Li et al. tracked the lane using a particle filter [27,28].

Various studies have been conducted on the lane detection using deep learning. Kim et al. detected lanes using convolutional neural networks [29]. Zou et al. detected lanes using convolutional neural networks and long short-term memory that is one of the recurrent neural networks [30]. Neven et al. detected lanes using LaneNet [31]. Ghafoorian et al. detected lanes using embedding loss driven generative adversarial networks [32].

Lane detection using deep learning has good performance but has a large amount of calculation. Therefore, the proposed method does not use deep learning-based lane detection but uses a general lane detection method to implement on an embedded platform.

The proposed method extracts lane features using combination line filters and region filters. The proposed method detects lines using inverse perspective mapping. The proposed method uses linear parabolic for the model. Furthermore, the proposed method uses a Kalman filter for tracking lanes.

Edge detection perceives drastic changes in the video pixel values of the input image. Based on this, edge detection finds the boundaries of streets and lanes. Changes in the image pixel values can be obtained by calculating the difference between the current and next pixels. Edge detection can be expressed as

$$E(x,y) = \sum_{y=1}^{H} \sum_{x=1}^{W} (I(x+1,y) - I(x-1),y) \tag{1}$$

where $E$ denotes the result of edge detection, $I$ denotes the input image, $W$ denotes the width of input image, and $H$ denotes the height of input image. Edge detection results are as shown in Figure 1.

Region detection is a method for detecting a region using the lane marking color. The region area is defined as follows: the image area turns a bright area into a dark area, and turns a dark area into a bright area. Region detection uses a dark-bright-dark function, and the filter for this function is as shown in Figure 2.

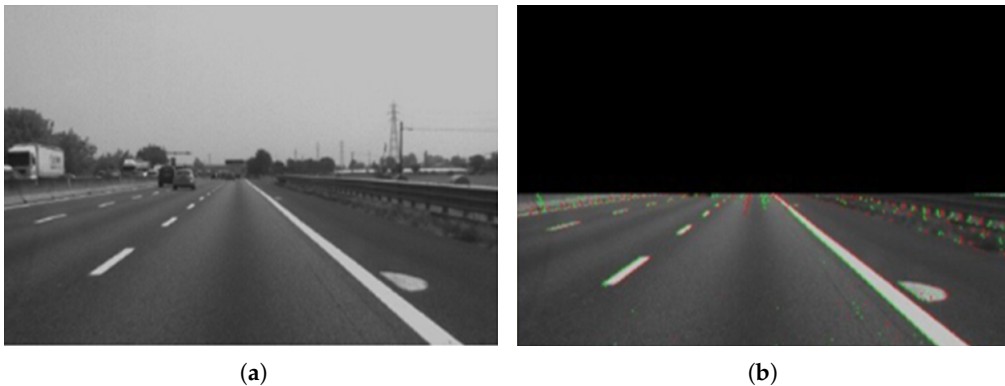

(**a**) (**b**)

**Figure 1.** Edge detection result: (**a**) input image, and (**b**) edge detection result.

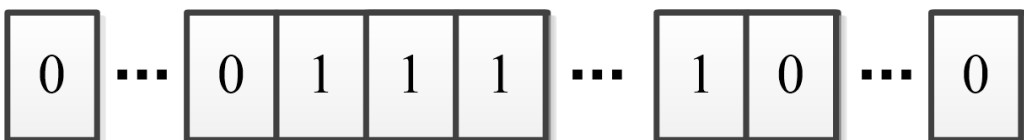

**Figure 2.** Filter for dark-bright-dark function.

The results of region detection using the dark-bright-dark function are shown in Figure 3. As can be seen, the lane width appears wider in the proximal region and narrower in the distant region of the input image. The actual lane area is calculated correctly, even when the lane width appears different along the input image, using various scales for the actual size of the black (0) and white (1) areas of the filter for the dark-bright-dark function.

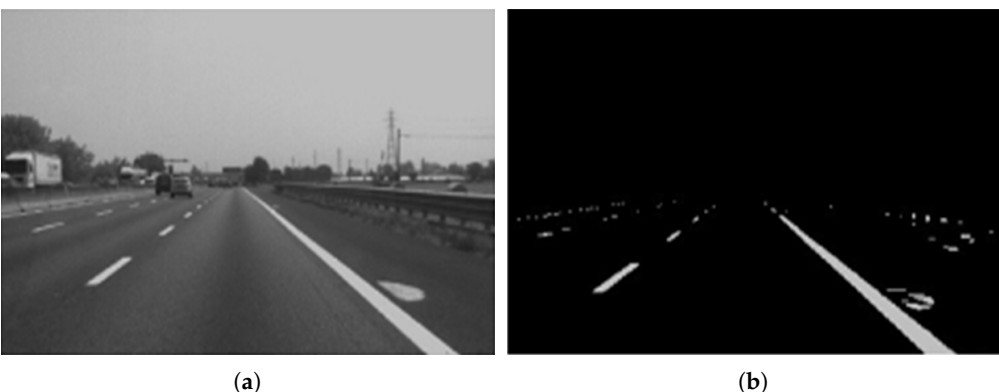

(**a**) (**b**)

**Figure 3.** Dark-bright-dark filter result: (**a**) input image, and (**b**) dark-bright-dark filter result.

It uses both edge and region detection results to more improve the accuracy of lane detection. To detect lanes, an edge detection algorithm calculates the median value between positive edge and negative edge from lane edges. In addition, edge detection calculates the median value from the region of interest (ROI) of a lane. It detects the lane if overlapped median values are detected from both the edge and region detection results. It also detects noise.

Figure 4 shows the ROI setting result using Kalman filter tracking and the line detection result using inverse perspective mapping. In Figure 4, the red lines are the line detection results, and the green lines are the ROI setting results.

Figure 5 shows the result of line fitting using a linear parabolic. In Figure 5, the green lines are the lane fitting results.

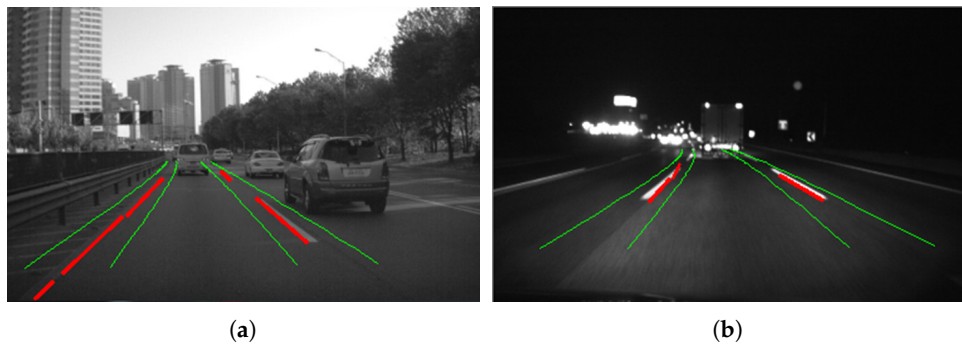

**Figure 4.** Line detection and lane tracking results: (**a**) daytime result, and (**b**) night result.

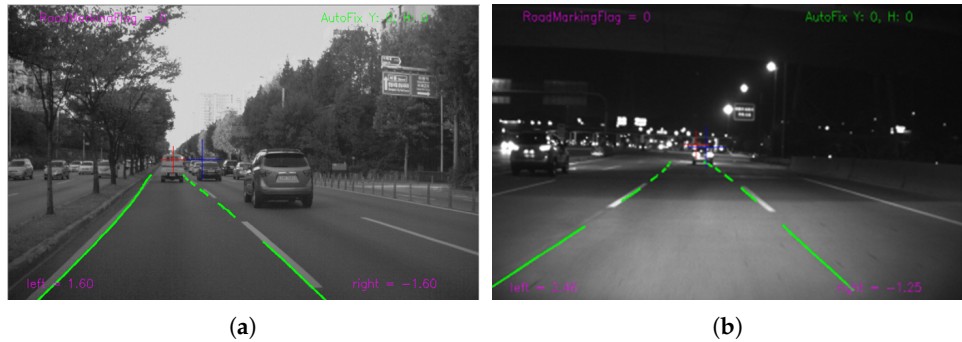

**Figure 5.** Line fitting results: (**a**) daytime result, and (**b**) night result.

Finally, the block diagram for the general lane detection algorithm with edge and region detection is shown in Figure 6. Subsequent to obtaining the input image from the front camera module, the lane's edge is detected and the lane region is extracted. Next, the lane component is extracted from the edge and region. Finally, the actual lane is selected and recognized by fitting, and the tracking algorithm is applied to enhance the accuracy of lane detection.

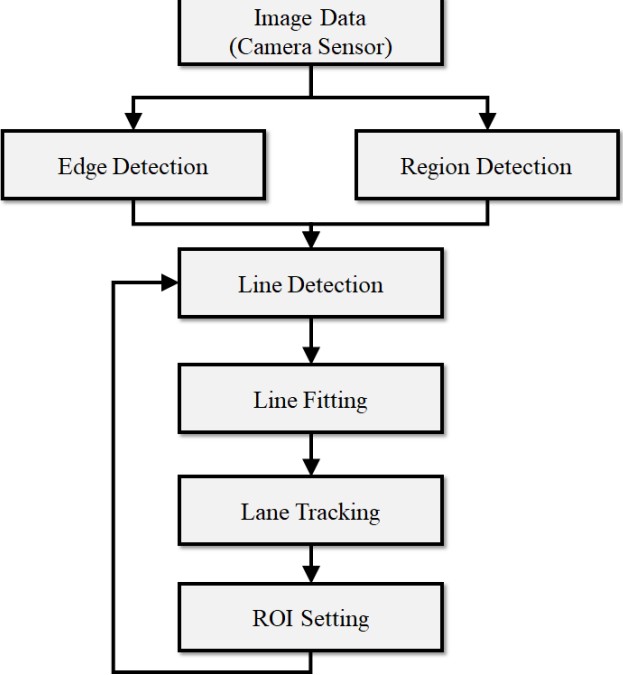

**Figure 6.** Block diagram for general lane detection algorithm.

### 2.2. Light Intensity Detection for Dual Light Sensor

Typically, solar sensors are used for air-conditioning systems to maintain the air temperature demanded by the driver. Dual light sensors integrate dual solar and twilight sensors into one chip. The circuit for the desired solar sensor angular response and for measuring the amount of light is shown in Figure 7. This circuit provides output currents from photodiodes that are proportional to the amount of light received and desired angular response [33].

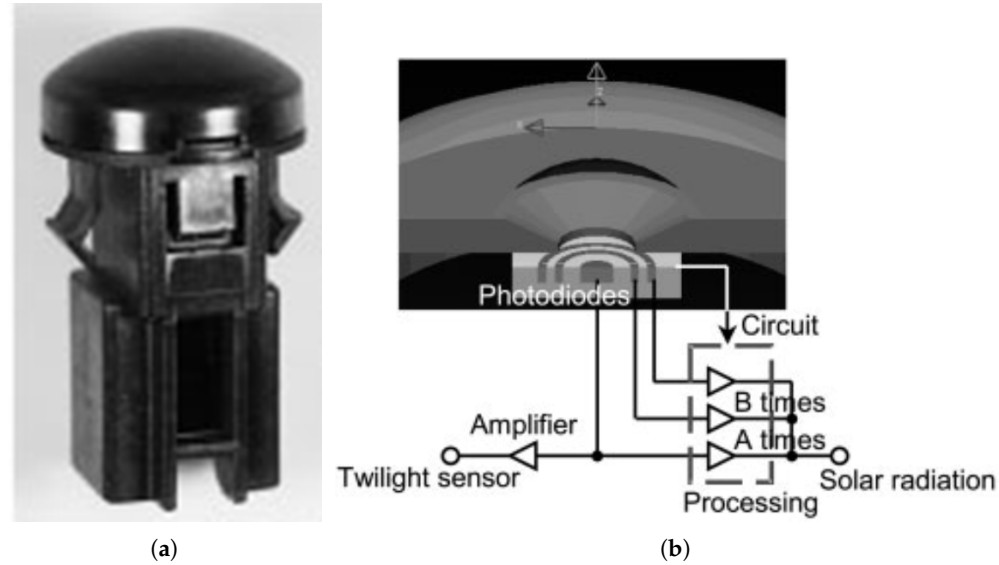

(**a**)  (**b**)

**Figure 7.** Dual light sensors: (**a**) dual light sensors, and (**b**) circuit for dual light sensors.

As a result, dual light sensors can cover a wide range of illumination, from twilight at only a few *lux* to full sunlight at 100,000 *lux*. Illumination data for dual light sensors are as shown in Figure 8a,b. The relative solar outputs for dual light sensors are as shown in Figure 8c [34]. The measured results for the amount of light in a plane surface are listed in Table 1. According to Figure 8 and Table 1, external conditions, such as backlight or night driving, can be recognized by the light density.

**Table 1.** Measured results for light amount in plane surface.

| Condition | Illumination (lux) |
|---|---|
| Sunlight | 107,527 |
| Full Daylight | 10,752 |
| Overcast Day | 1075 |
| Very Dark Day | 107 |
| Twilight | 10.8 |
| Deep Twilight | 1.08 |
| Full Moon | 0.108 |
| Quarter Moon | 0.0108 |
| Starlight | 0.0011 |
| Overcast Night | 0.0001 |

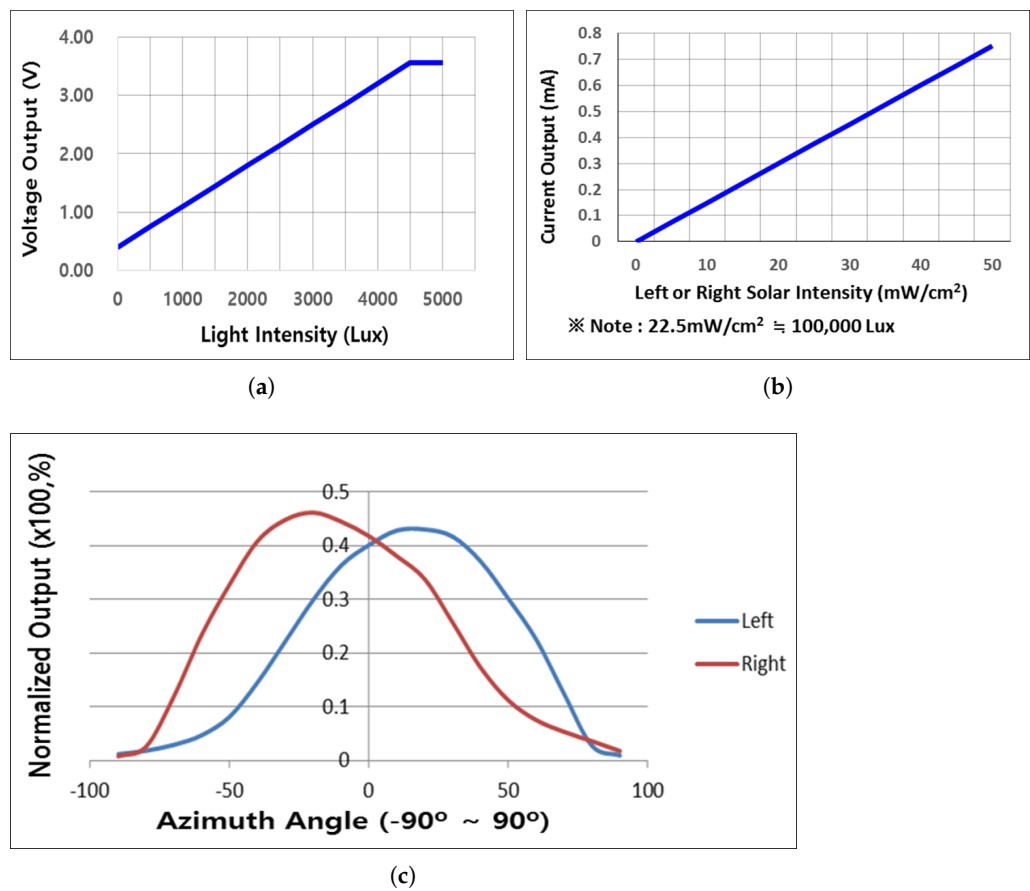

**Figure 8.** Illumination for dual light sensor: (**a**) twilight sensor voltage output vs. light level, (**b**) solar sensor current output vs. light level, and (**c**) relative solar output for dual light sensors.

### 3. Proposed Lane Detection Method with Illumination Information

The proposed lane detection method uses illumination intensity information in order to improve those situations where the lane edges in an image have relatively weak contrast or where there are strong distracting edges.

#### 3.1. Block Diagram for Proposed Lane Detection Method

The dual light sensor provides information about the amount and direction of light. As shown in Table 1, weather conditions are extracted with illumination information. In addition, backlight and side-light are detected using the direction of the light. Tunnel information is extracted with rapidly changing lighting.

The proposed method improves lane detection by using information from the dual light sensor. In the proposed method, three parts are improved in lane detection: (1) image quality, (2) edge detection, and (3) lane tracking.

Figure 9 shows the block diagram of the proposed method. As shown in Figure 9, it is configured to receive input image and dual light sensor data simultaneously. The red line indicates the improved part using the dual light sensor compared to the lane recognition method described in Section 2.

Image quality enhancement uses night, day, and sun angle information from dual light sensors. In the proposed method, the enhanced image is defined as second image data. Second image data create an image with the best quality for lane detection. Edge detection improvement changes the threshold using tunnel information and twilight information from the dual light sensor. If the threshold is changed, the edge of the lane is extracted even in the image where the edge of the lane does not appear well. Lane tracking improvement narrows the ROI by using tunnel and side-light information from a dual light sensor. If

the ROI is reduced, false lane features appearing inside the tunnel or due to shadows are removed.

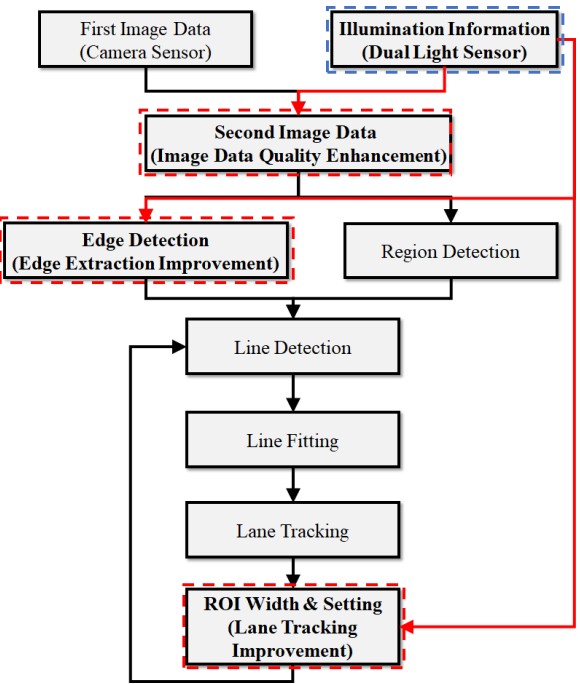

**Figure 9.** Block diagram of the proposed method.

### 3.2. Image Data Quality Enhancements

General lane detection algorithms require the control of exposure time or varying of analog gain from image sensor information. However, image sensors cannot detect external conditions, such as backlight or night driving situations. As a result, gain and exposure control response time are delayed, thus leading to wrong detection and a decrease in the lane detection rate. Therefore, enhanced image processing that uses an additional dual light sensor is applied for the recognition of backlight conditions or night driving situations and enhancement of the gain and exposure control performance (faster response time).

Exposure time can be calculated with Equations (2) and (3). The value of $E_I$ is calculated by changing $C_E$ using the light sensor illumination from $E_O$; $C_E$ can be calculated by the weighted sum of $L$ and $A$ and is directly proportional to the log value of $L$ and inversely proportional to the absolute value of $A$. Similarly to Equations (2) and (3), analog gain can be calculated with Equations (4) and (5). The value of $G_I$ is calculated by changing $C_G$ using the light sensor illumination from $G_O$.

$$E_I = (1 - C_E) \times E_O \tag{2}$$

$$C_E = \omega_{LE} \log_2 L + \omega_{AE}(\frac{\pi}{2} - |A|) \tag{3}$$

$$G_I = (1 - C_G) \times G_O \tag{4}$$

$$C_G = \omega_{LG} \log_2 L + \omega_{AG}(\frac{\pi}{2} - |A|) \tag{5}$$

where $E_I$ denotes the next exposure time, $C_E$ denotes the constant of the control of exposure time by the illumination value, and $E_O$ denotes the current exposure time of the camera sensor; $G_I$ denotes the next gain, $C_G$ denotes the constant of the gain control by the illumination value, $G_O$ denotes the current gain of the camera sensor, $L$ denotes the illumination value, which is derived from the information of the dual light sensor, and $A$ denotes the angle of sun, which is derived from the information of the dual light sensor.

### 3.3. Edge Detection Improvement

Lane detection performance is related to edge and region detection. If the threshold for edge and region detection is too high, the lane detection rate decreases. On the contrary, if the threshold is too low, the lane detection quality is lower and the false recognition rate increases because of the poor image quality. In this paper, the light intensity information from dual light sensors is used for various thresholds of edge and region detection, which enhances the quality of lane detection on critical conditions, such as tunnel and twilight driving.

Figure 10 shows the tunnel and twilight images taken using the front camera. As shown in Figure 10, the edge and region of lanes markings do not appear clearly in the tunnels and at twilight.

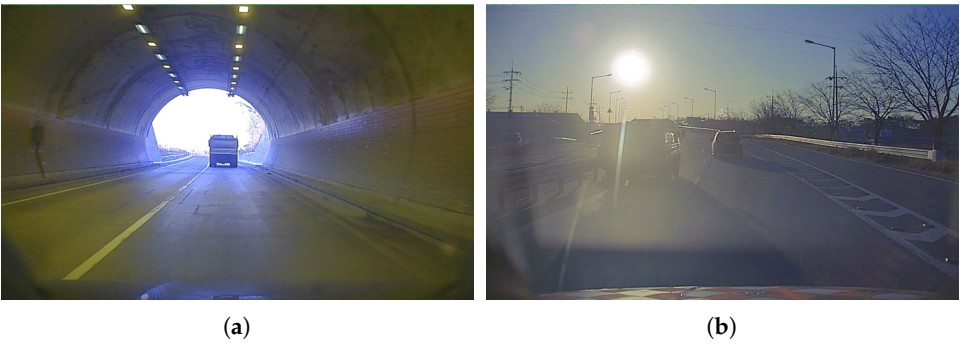

| (a) | (b) |

**Figure 10.** Examples of tunnel and twilight image: (**a**) tunnel image, and (**b**) twilight image.

The control for the threshold of edge and region detection can be calculated with Equation (6).

$$Th_E = \begin{cases} Th_L, & \text{if } \text{Tunnel=1 or Twilight=1} \\ Th_H, & \text{else} \end{cases} \tag{6}$$

where $Th_E$ denotes the result of the threshold value for edge and region detection, and $Th_L$ and $Th_H$ denote the low and high threshold values, respectively.

Figure 11 shows the edge detection result from the tunnel image. Figure 11a shows the edge detection result when the threshold is high, and Figure 11b shows the edge detection result when the threshold is low. As shown in Figure 11, when the threshold is high in the tunnel image, the edges of the lane markings are not visible. In contrast, when the threshold is low, the edges of the lane markings are visible.

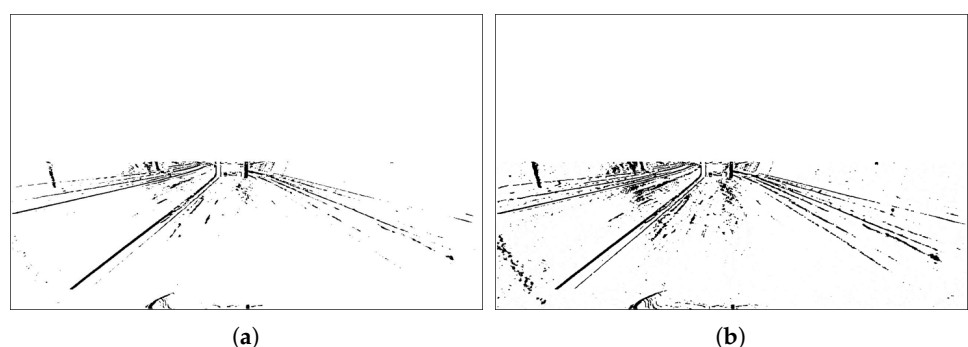

| (a) | (b) |

**Figure 11.** Edge detection results: (**a**) result ($Th_E = Th_H$), and (**b**) result ($Th_E = Th_L$).

### 3.4. Lane Tracking Improvement

External illumination information is efficient not only for edge and region detection, but also for tracking performance in specific cases, such as when entering tunnels or detecting guardrail shadows caused by lateral light sources.

Figure 12 shows that a line similar to the lane marking is created by the shadow from the inside of the tunnel and side light. Figure 12a is an image of the inside of a tunnel, and Figure 12b shows a shadow image of a guardrail with side lighting.

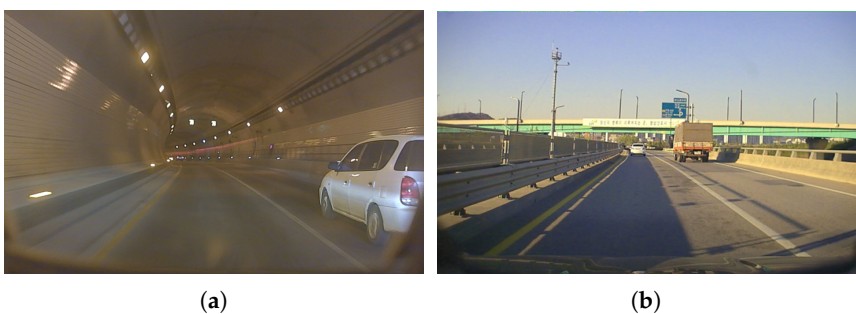

(**a**)　　　　　　　　　　　　　　(**b**)

**Figure 12.** Examples of tunnel and guardrail with side lighting image: (**a**) tunnel image, and (**b**) guardrail with side lighting image.

The proposed method uses a third polynomial model based on vehicle coordinates. Figure 13 shows the lane marking model. As shown in Figure 13, the proposed method uses a third polynomial model in which the longitudinal direction of the vehicle is the *x*-axis [35].

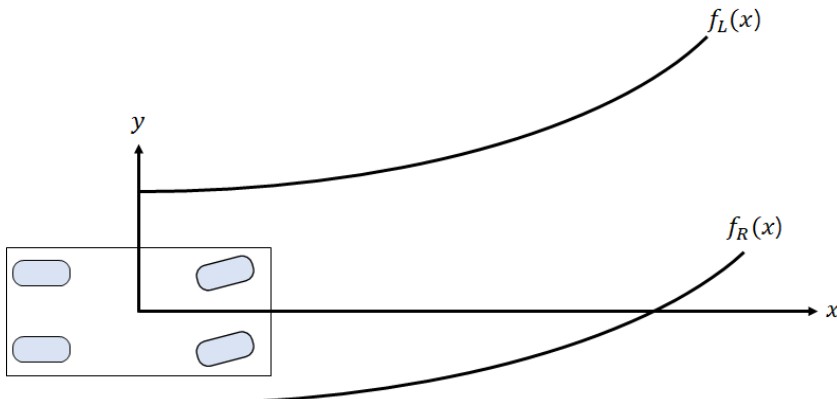

**Figure 13.** Example of lane marking model.

Lane marking model $f_L(x)$ and $f_R(x)$ are defined by Equation (7) [35].

$$f_L(x) = a_L \cdot x^3 + b_L \cdot x^2 + c_L \cdot x + d_L$$
$$f_R(x) = a_R \cdot x^3 + b_R \cdot x^2 + c_R \cdot x + d_R \tag{7}$$

where $a_L$ and $a_R$ represent curvature rate, $b_L$ and $b_R$ are curvature, $c_L$ and $c_R$ are heading angle, and $d_L$ and $d_R$ are offset of left and right lanes, respectively.

The control for tracking the area size can be calculated with Equation (8). The Kalman filter is used for tracking, and the position of the next lane is estimated by Kalman prediction [36–38]. The lane position for the next frame is estimated through the relationship between the estimated and current lane positions.

$$W_a = \alpha W_k \begin{cases} \alpha = 0.5, & \text{if } \text{Tunnel}=1 \text{ or Sidelight}=1 \\ \alpha = 1, & \text{else} \end{cases} \tag{8}$$

where $W_k$ is the range of the area calculated by the Kalman filter prediction, and $W_a$ is the estimation range when $\alpha$ is 0.5 if the lateral light or tunnel-entering condition, or $\alpha$ is 1 for all other cases. $W_k$ is calculated by Equation (9).

$$W_k = (d_C - d_{Kp}) + C_d. \tag{9}$$

where $d_C$ is offset of lane marking model in current frame, $d_{Kp}$ is offset of Kalman prediction result, and $C_d$ is constant value of detection range.

Figure 14 shows the calculation tracking range result. The green lines are lane detection results in the current frame, and the red lines are the calculated tracking range. Figure 14a shows the result with a narrow tracking range, and Figure 14b shows the result with a normal tracking range.

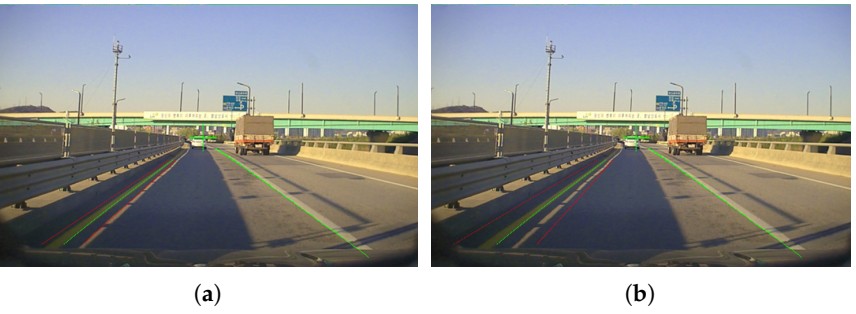

(**a**)                                                    (**b**)

**Figure 14.** Examples of calculated tracking range result: (**a**) narrow tracking range result, and (**b**) normal tracking range result.

## 4. Experimental Results

### 4.1. Design Results for Proposed Integrated Camera with Dual Light Sensor

The configuration of the integrated front camera module and dual light sensor is shown in Figure 15. The front camera module receives an image from the external vehicle environment with an image sensor. The input image is optimized for quality using image signal processing. The vision processor executes the lane recognition algorithm. The dual light sensor detects the intensity of the sun on the right and left regions with dual solar and twilight sensors and estimates the position of the sun and light density. A photograph of the realized integrated front camera module with dual light sensor is shown in Figure 16. The designed circuit board and appearance are shown in Figure 16a,b, respectively. Because the developed camera has a highly integrated size, it has advantages such as reduced camera and sensor size and cost-effectiveness.

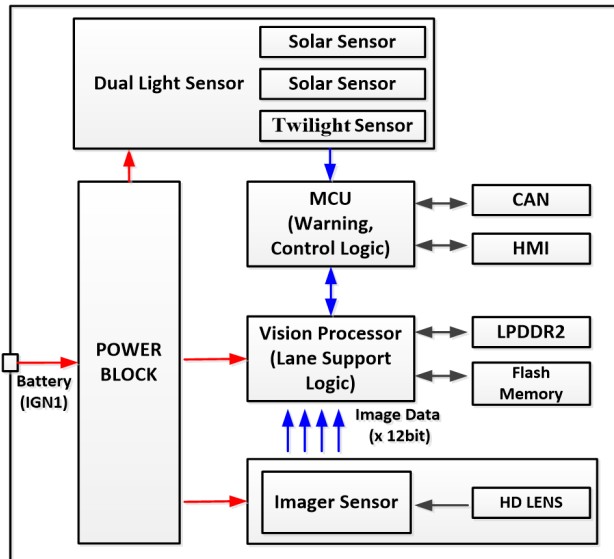

**Figure 15.** Actual design for circuit board of proposed integrated front camera module dual light sensor.

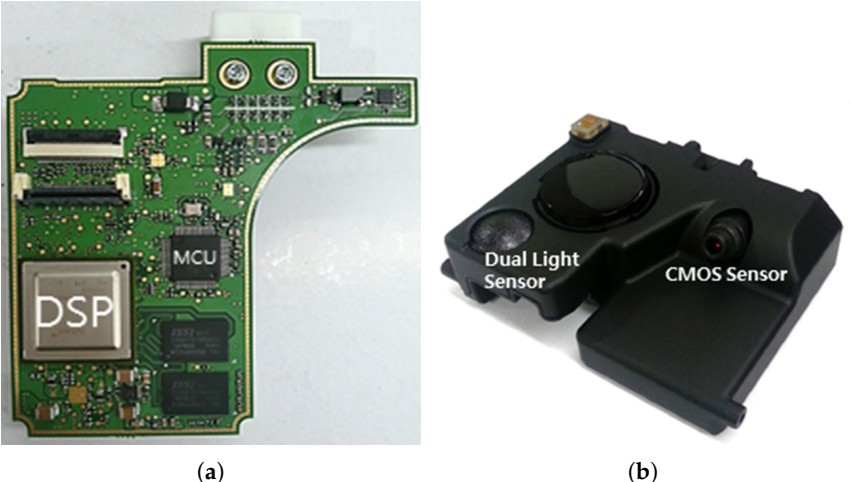

(**a**)　　　　　　　　　　　　　　　　　　　　　　　(**b**)

**Figure 16.** Photograph of realized integrated front camera module with dual light sensor: (**a**) designed circuit board and (**b**) appearance.

The specifications for the complementary metal oxide semiconductor (CMOS) and dual light sensors are listed in Table 2. The camera can detect the lane up to 90 m for the lane support system. The dual light sensor can measure light intensity from 0 lux to 100,000 lux.

**Table 2.** CMOS sensor and dual light sensor.

| Item | Parameter | Specification |
|---|---|---|
| CMOS Sensor | FOV (Field of view) | 52° (H) × 38 ° |
| | Resolution | 1280 × 800 (HD) |
| | Frame rate | 30 fps |
| | Dynamic range | 115 dB |
| | Detection range | 90 m |
| Dual Light Sensor | Light intensity | 0 to 100,000 lux |
| | Sensor output current | 145 mA ± 15% |
| | Angular response (elevation angle) | −90°/90° |
| | Angular response (azimuth) | 40° |

*4.2. Experiment Results*

The environment for the lane support system tests is shown in Figure 17. The vehicle is installed with an integrated front camera module in front of the windshield. These tests are performed in dry conditions with an ambient temperature of approximately 25 °C.

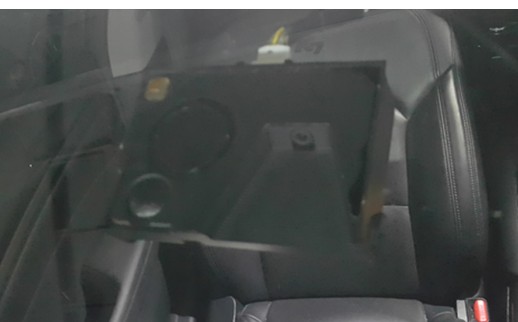

**Figure 17.** Integrated front camera module installed on inner windshield.

The test results for the enhancement of image data quality are shown in Figure 18. Figure 18a shows the original image data for the backlight condition. The image data are saturated with white because of sunlight, and thus it is difficult to distinguish the

lane on the road surface. In contrast, the results after enhancing the image quality for the backlight condition are shown in Figure 18b. Based on Figure 8b and Table 1, we applied the sunlight condition to Equations (2) and (3). As a result, the lane image data shows a clearly distinguished lane within the surrounding road surface.

Figure 18c shows the original image data for the night driving situation. In some areas, the image data are saturated by other light sources (such as other vehicles' headlight and backlight, streetlights, and neon signs), and thus it is difficult to distinguish the lane on the road surface. On the contrary, the results after enhancing the image quality for the night driving situation are shown in Figure 18d. Based on Figure 8a and Table 1, we applied the twilight condition to Equations (2) and (3). Figure 18e shows the original image data for the entering tunnel situation. In some areas, the image data are saturated with black because of shadow, and thus it is difficult to distinguish the lane on the road surface. In contrast, the results after enhancing the image quality for the entering tunnel situation are shown in Figure 18f. Based on Figure 8a and Table 1, we applied the twilight condition to Equation (2) and (3). As a result, the lane image data show a clearly distinguished lane within the surrounding road surface. Finally, we obtain optimized image data in the worst environment conditions using the light density from the dual light sensor for the control of exposure time and gain.

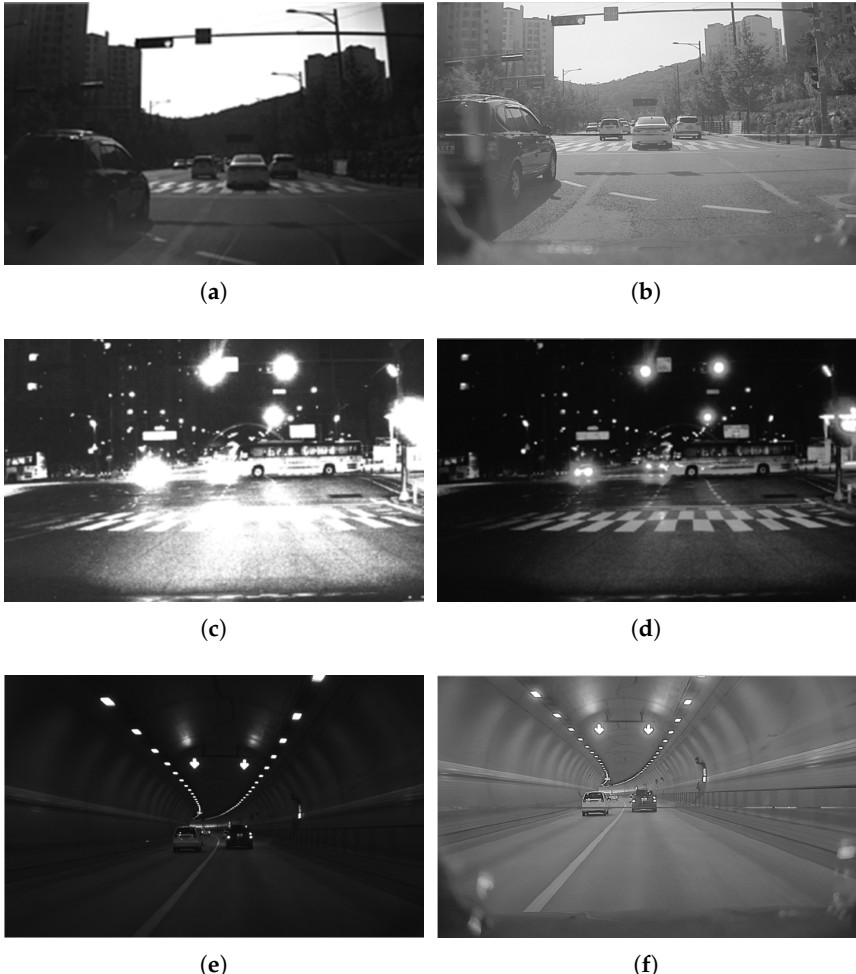

(a)                    (b)

(c)                    (d)

(e)                    (f)

**Figure 18.** Test results for image quality enhancements: (**a**) backlight condition (without illumination data), (**b**) backlight condition (with illumination data), (**c**) night driving situation (without illumination data), (**d**) night driving situation (with illumination data) (**e**) entering tunnel situation (without illumination data), and (**f**) entering tunnel situation (with illumination data).

The test results for the improvement of edge detection and the tracking process are shown in Figure 19a–d. Figure 19a shows the original image data when entering a tunnel. Because of the sudden changes in the external light condition, the lane detection algorithm perceives the left lane incorrectly. The results of the tracking process improvement when entering the tunnel are shown in Figure 19b. Based on Figure 8a and Table 1, we applied the tunnel condition to Equation (5). Table 1 lists twilight as measuring 10.8 *lux*; this illumination value is applied because the condition is similar to entering a tunnel. As a result, we can confirm correct detection of the left lane, as shown in Figure 19b.

Figure 19c shows the original image data from the lateral light condition. Because of the guardrail shadow, the front camera module cannot detect the left lane. The results of improving edge detection under lateral light conditions are shown in Figure 19d. Based on Figure 9, we can obtain the normalized output, which is less than 0.5 for the lateral light condition where the sunlight appears at less than 30° from the horizon. We applied the lateral light condition to Equation (4). As a result, we can correctly detect the left lane.

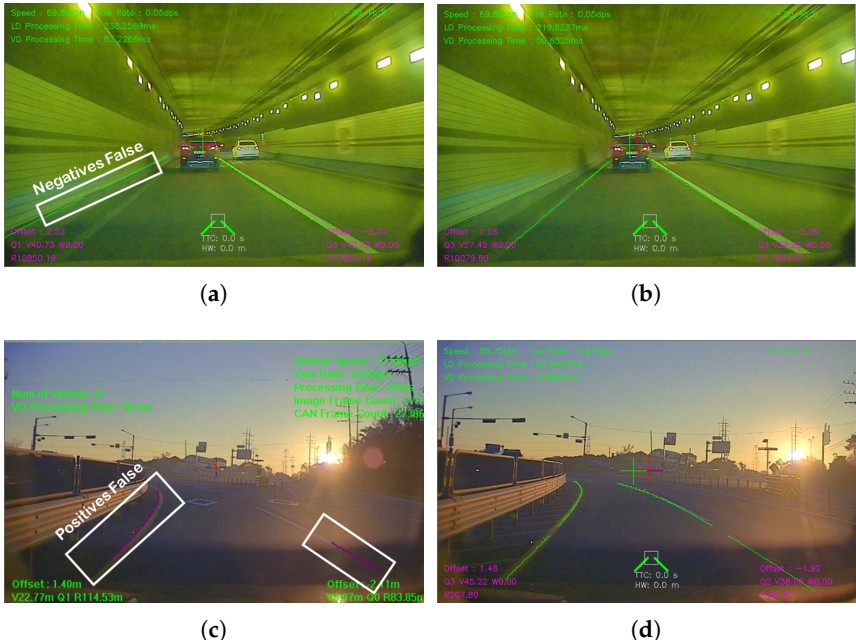

**Figure 19.** Test results for edge detection and tracking process improvements: (**a**) entering tunnel condition (without illumination data), (**b**) entering tunnel condition (with illumination data), (**c**) side-light driving situation (without illumination data), and (**d**) side-light driving situation (with illumination data).

Finally, we can obtain the optimized lane detection results without incorrect detection or recognition in the worst environment conditions using the light density from the dual light sensor for the control of the threshold of the edge (region) area and tracking performance.

In order to obtain a more precise assessment, tests were conducted under various conditions on a real road. Typically, assessments of lane support systems evaluate the rate of correctly recognizing the lane and the false alarm rate. For our test, we equipped a vehicle with a camera in order to detect the accuracy of our method and the number of false alarms. We drove the test vehicle for a total of 728 km [39].

First, real-road tests were conducted under various external situations: day and night driving, backlight conditions, and the presence of lateral light, rain, and snowfall. The analysis indicates good results in the cases of day, night, backlight, and lateral light, as shown in Figure 20a,b.

Second, real-road tests were performed under different lane types: those with a solid line, dashed line, curved line, road marking line, and crosswalk. The analysis indicates

good results in the cases of solid line, dashed line, curved line, and road marking line, as shown in Figure 21a–e.

For each performance index, the number of correct detections, false positives, and false negatives was evaluated. A total of 59 cases were correctly detected in the first set of tests. Only one false negative is present (caused by the heavy rain condition, as shown in Figure 20a). CMOS camera technology cannot compete against the human eye. Under heavy rain conditions, lane detection is not possible.

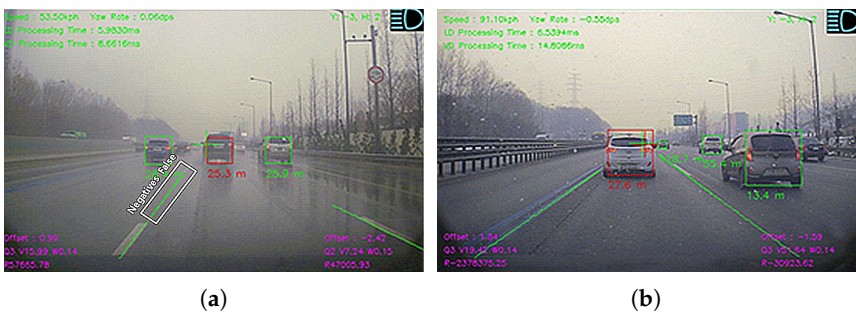

(a)       (b)

**Figure 20.** Real-road tests were conducted under various external situations: (**a**) heavy rain, and (**b**) snowfall.

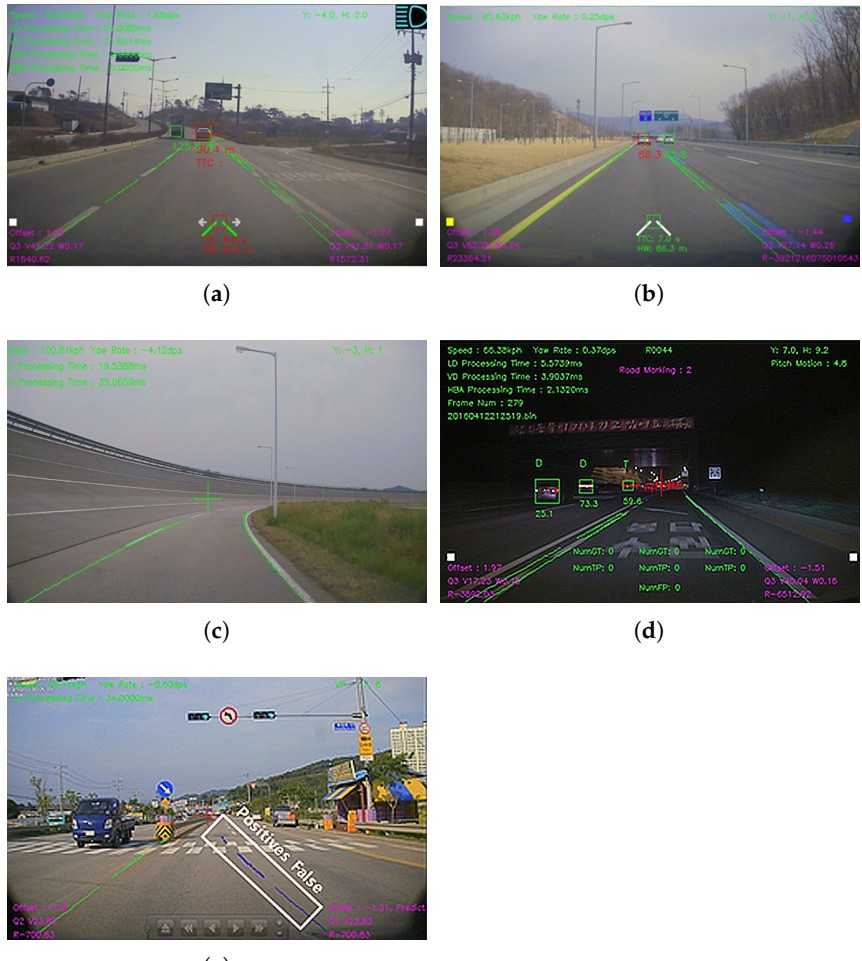

**Figure 21.** Real-road tests were performed under different lane types: (**a**) dashed line + dashed line, (**b**) solid line + dashed line, (**c**) curved line, (**d**) road marking line, and (**e**) crosswalk.

A total of 36 cases were correctly detected in the second set of tests. Two false negatives were present (caused by the crosswalk, as shown in Figure 21e). There are limitations in the proposed method. The results of the real situation tests are listed in Table 3.

**Table 3.** Test results from real road.

| Perform. Index | Correct Detection | False Negatives | False Positives |
|---|---|---|---|
| External situations | 59 | 1 | 0 |
| Different lane types | 36 | 0 | 1 |

## 5. Conclusions

In the near future, vehicles will have a variety of ADAS for autonomous driving. Lane support systems are part of the safety assistance systems and are the foundation for achieving this goal.

In this paper, in order to improve those cases where the lane edges in an image have relatively weak contrast, or where there are strong distracting edges, we proposed a novel lane detection method that uses illumination intensity information. We presented a new algorithm for enhancing the quality of the image data and improving edge detection and lane tracking using the illumination information. For our evaluation, we designed an integrated front camera module with dual light sensor and mounted it on the windshield of a vehicle. Lane detection performance was measured under the worst conditions. For comparing the results before and after applying the proposed method, a total of 105 cases were correctly detected in the tests (external situations and different lane types). Only one false positive (caused by rainy conditions) and two false negatives (caused by crosswalk and road marking conditions) were found. Finally, when considering the results of various evaluation tests, we could confirm the performance improvement of the lane support system.

Future work will be extended to investigate enhancements for lane detection, including recognizing a variety of lane types (crosswalks, road markings, and so on), in order to reduce the number of false negatives. We also plan to improve deep learning-based lane detection using dual light sensors.

**Author Contributions:** Y.L. developed the algorithm and performed experiments. M.-k.P. developed the H/W system, developed the algorithm, and performed experiments. M.P. contributed to the development of the algorithm, validation of experiments, and research supervision. Y.L. and M.-k.P. contributed to writing the manuscript. M.P. contributed to reviewing and editing the manuscript. All authors have read and agreed to the published version of the manuscript.

**Funding:** This research was supported by Basic Science Research Program through the National Research Foundation of Korea(NRF) funded by the Ministry of Education (2018R1D1A1B07048143153038 2068210105). This work is supported by the Korea Agency for Infrastructure Technology Advancement (KAIA) grant funded by the Ministry of Land, Infrastructure and Transport (grant no. 22AMDP-C160501-02). Also, This work was supported by the Ministry of Trade, Industry and Energy(MOTIE, Korea) (No. 20018055, Development of fail operation technology in Lv.4 autonomous driving systems).

**Conflicts of Interest:** The authors declare no conflict of interest.

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
