# Peer review of "Improving Lane Detection Performance for Autonomous Vehicle Integrating Camera with Dual Light Sensors"

_electronics, doi:10.3390/electronics11091474_

Round 1
Reviewer 1 Report
In this paper, an integrated camera with dual light sensors is developed to improve lane detection performance under various conditions. An efficient algorithm is proposed to enhance the image quality and improve the edge extraction and lane tracking performance using illumination information. Here are some comments:
(1)There exists an error in Figure 16 (c), which is the same as Figure 16 (e). Please check!
(2)Recently,deep learning algorithm has made great progress in the field of lane detection. Can the proposed system support deep learning algorithms?
Author Response
Dear Reviewer
Thank you for your review again.
We are uploading our point-by-point response to the comments that included (a) an updated manuscript with indicating changes, and (b) a clean updated manuscript without highlights (PDF main document).
Thank you.
Sincerely,
Yun Hee Lee

Reviewer 2 Report
The research paper presents an interesting overview of dual light sensors to improve safety. However, the overall methodology needs to be improved. At the moment is a collection of figures and charts trying to explain the process. But, this section needs to be more cohesive and better explained. It is recommended that a step-by-step development. This needs to be clear and deliberated comprehensively. Also, it will be good to include more recent references to highlight the gap found within the literature. Finally, there are some minor grammar issues that require some attention.
Author Response

(The authors gave the same response as above.)
